# Safety and Tolerability of the BNT162b2 mRNA COVID-19 Vaccine in Dialyzed Patients. COViNEPH Project

**DOI:** 10.3390/medicina57070732

**Published:** 2021-07-19

**Authors:** Karolina Polewska, Piotr Tylicki, Bogdan Biedunkiewicz, Angelika Rucińska, Aleksandra Szydłowska, Alicja Kubanek, Iwona Rosenberg, Sylwia Rodak, Waldemar Ślizień, Marcin Renke, Alicja Dębska-Ślizień, Leszek Tylicki

**Affiliations:** 1Department of Nephrology Transplantology and Internal Medicine, Medical University of Gdansk, 80-210 Gdańsk, Poland; kpolewska@gumed.edu.pl (K.P.); ptylicki@gumed.edu.pl (P.T.); adeb@gumed.edu.pl (A.D.-Ś.); leszek.tylicki@gumed.edu.pl (L.T.); 2Department of Occupational, Metabolic and Internal Diseases, Medical University of Gdansk, 80-210 Gdańsk, Poland; ang.rucinska@gmail.com (A.R.); aleksandra.szydl@gmail.com (A.S.); alicja.kubanek@gumed.edu.pl (A.K.); mrenke@gumed.edu.pl (M.R.); 3NZOZ Diaverum, 81-519 Gdynia, Poland; rosenberg.iwona@gmail.com (I.R.); sylwia.rodak@diaverum.com (S.R.); waldemar.slizien@diaverum.com (W.Ś.)

**Keywords:** COVID-19, SARS-CoV-2, vaccine, dialysis

## Abstract

*Background and Objectives*: The Pfizer-BioNTech (BNT162b2) COVID-19 mRNA vaccine has demonstrated excellent efficacy and safety in phase 3 trials. However, no dialyzed patients were included, and therefore safety data for this patient group is lacking. The aim of the study was to assess the safety and tolerances of vaccinations with BNT162b2 performed in chronically dialyzed patients. *Materials and Methods*: We performed a prospective cohort study including a group of 190 dialyzed patients (65% male) at median age 68.0 (55–74) years. 169 (89.0%) patients were treated with hemodialysis and 21 (11.0%) with peritoneal dialysis. The control group consisted of 160 people (61% male) without chronic kidney disease at median age 63 (range 53–77) years. Both groups were vaccinated with BNT162b2 with a 21-day interval between the first and the second dose. Solicited local and systemic reactogenicity, unsolicited adverse events and antipyretic and pain medication use were assessed with a standardized questionnaire. The toxicity grading scales were derived from the FDA Center for Biologics Evaluation and Research guidelines. *Results*: 59.8% (dose 1), 61.4% (dose 2) and 15.9% (dose 1), 29.4% (dose 2) dialyzed patients reported at least one local and one systemic reaction respectively within seven days after the vaccination. Many local and systemic solicited reactions were observed less frequently in dialyzed patients than in the age and sex matched control group and much less frequently than reported in the pivotal study. They were mostly mild to moderate, short-lived, and more frequently reported in younger individuals and women. No related unsolicited adverse events were observed. *Conclusions*: We have shown here that BNT162b2, an mRNA vaccine from Pfizer-BioNTech against SARS-COV-2 is safe and well-tolerated by dialyzed patients. The results can be useful for the nephrological community to resolve patients’ doubts and reduce their vaccine hesitancy.

## 1. Introduction

The devastation of the COVID-19 (coronavirus disease 2019) pandemic has been rippling through the population of dialyzed patients. Previous studies have reported horrifyingly disproportionate age-adjusted rates of cases, with fatality rates varying from 16% to 32% deaths [1,2]. In the absence of an effective COVID-19 treatment, vaccination is the only chance to improve the extremely poor prognosis in this population. In a nationwide vaccine acceptability survey performed in 150 dialysis units in the United States a significant proportion of hemodialyzed (HD) patients hesitate to vaccinate against COVID-19 for fear of potential side effects. Hesitancy among people overall may have recently increased, at least for the short term, because of concerns over rare thrombotic events among people who receive certain types of COVID-19 vaccine [3,4]. BNT162b2 mRNA vaccine against SARS-CoV-2 (severe acute respiratory syndrome coronavirus 2) approved for use in the US and Europe was shown to have a favorable safety profile, and its reactogenicity was generally mild or moderate [5]. However, pivotal studies were not performed in the population of patients on dialysis maintenance. To expand on this issue, we performed a study to assess the safety and tolerances of vaccinations with BNT162b2 performed in chronically dialyzed patients. Being aware of what to expect after vaccination against COVID-19 may help educate this population, dispel false information, and reduce vaccine hesitancy [6].

## 2. Materials and Methods

The prospective observational study was performed in all chronically dialyzed patients from the Department of Nephrology Transplantology and Internal Medicine, Medical University of Gdansk and NZOZ Diaverum Hemodialysis Unit in Gdynia. Patients met the study inclusion criteria if they were on chronic dialysis for at least 1 month and had received intramuscularly two-dose vaccination with mRNA BNT162b2 vaccine (BionTech/Pfizer Comirnaty) with a 3-week interval between the first and the second doses in the period from 25 January 2021 to 18 March 2021 according to the rules of the national immunization program and the manufacturer’s recommendations. HD patients were vaccinated on the day of dialysis after HD session. The control group were people without chronic kidney disease matched for age and gender, vaccinated in the same period in the University Centre of Maritime and Tropical Medicine in Gdynia with the same vaccine and doses. In this report, safety data is reported for all participants who provided informed consent and received at least one dose of the vaccine. Medical data of patients from both groups was extracted from their medical records.

The primary end points of the study were solicited common and expected adverse reactions shortly following vaccination (reactogenicity), use of antipyretic or pain medications and unsolicited adverse events and serious adverse events, i.e., those reported by the participants without prompts from the medical staff or observed by their physicians through 1 month after the second dose. Reactogenicity assessments included solicited injection (local) site reactions (pain, redness, swelling) and systemic reactions (fever, fatigue, headache, chills, vomiting, diarrhea, new or worsened muscle pain, and new or worsened joint pain). Data was obtained through triple phone interviews performed by health staff according to the standardized questionnaire, 7 days after the first and the second dose, and 30 days after the final vaccination (Appendix A). The grading scales (the same as in the pivotal trial) used in this study were derived from the FDA Center for Biologics Evaluation and Research (CBER) guidelines on toxicity grading scales for healthy adult volunteers enrolled in preventive vaccine clinical trials [5]. Serious adverse events were defined as any untoward medical occurrence that resulted in death, was life-threatening, required inpatient hospitalization or prolongation of existing hospitalization, or resulted in persistent disability/incapacity. Assessment of the causality of adverse events following vaccination was done by two physician using the WHO algorithms [7], and were finally verified by the senior investigator. In secondary analyses we evaluated the vaccine safety in subgroups. We used the following strata: age (≤55 years vs. >55 years, the same as in the pivotal study), sex, comorbidities (binary variable, with or without comorbidities), smoking habit (binary variable), dialysis modality (hemodialysis vs. peritoneal dialysis) and BMI (body mass index), Charlson Comorbidity Index (CCI) and duration of dialysis treatment with median as the threshold. The study is part of the ‘COVID-19 in Nephrology’ (COViNEPH) project registered in the ClinicalTrials.gov, identifier NCT04905862.

We report descriptive results of safety analyses, and the sample size was not determined on statistical hypothesis testing. Data was presented as a percent for categorical variables and median (interquartile range; IQR) for continuous variables. Chi-square or Fisher’s exact test was used for categorical variables. Mann-Whitney U-tests were used to compare continuous variables. Univariable and multivariable logistic regression analyses were adopted to identify risk factors associated with reactogenicity of dialyzed patients. All variables from the univariable analysis with a *p* value < 0.1 were entered into a bidirectional-stepwise multivariable logistic regression analysis. *p* < 0.05 was considered significant.

## 3. Results

### 3.1. Patients

Between 25 January 2021 and 18 March 2021, a total of 206 of dialyzed patients from both units were vaccinated with BNT162b2. Sixteen people refused to participate in the study. Finally, 190 patients, (123 men, 64.7%) with a median age (interquartile range; IQR) 68.0 (55–74) years, median duration of dialysis treatment of 36 (14–67) months and median comorbidity index of 6 (4–8) were enrolled in the study. Then, 169 (89.0%) patients were treated with in-center hemodialysis and 21 (11.0%) with peritoneal dialysis. The most common cause of end-stage renal disease was diabetes. The control group consisted of 160 people (61% male) at median age 63 (53–77) years without chronic kidney disease. Dialysis patients (the study group) had a significantly higher Charlson Comorbidity Index, lower BMI and they suffered significantly more often from diabetes and arterial hypertension compared to the control group (Table 1).

### 3.2. Local Reactogenicity

Of the dialyzed patients, 59.8% and 61.4% reported at least one local site reaction within 7 days after the first and the second injection of BNT162b2, respectively. They reported mostly mild-to-moderate injection-site reactions. Three patients (1.6%) had severe local symptoms. No grade 4 local reactions were reported. Pain at the injection-site was the most frequent local reaction among vaccinated. A similar proportion of dialyzed patients reported any solicited local reactions compared with control recipients but less than reported in phase 3 trials [5]. Swelling and redness were reported significantly less by the dialyzed than the controls after the first and the second dose. The median delay in the onset of local reactions and its median duration were 1 day and 2.5 days, respectively. Details are presented in Figure 1 and Appendix A.

### 3.3. Systemic Reactogenicity

At least one of solicited systemic reactions occurred in 15.9% of the dialyzed patients after the first dose. Similar to the control group, they were more frequently reported after the second dose of BNT162b2 (29.4%; *p* < 0.01). Fatigue followed by muscle pains, joint pains and headaches were the most frequent solicited systemic reactions in both groups. A majority of the dialyzed patients reported only mild-to-moderate systemic reactions. No grade 4 systemic reactions were reported. A similar proportion of the dialyzed patients reported any solicited systemic reactions compared with control age and sex matched recipients, but much less than reported in the phase 3 trials [5]. New or worsened muscle pains and headaches after the second dose of BNT162b2 were reported significantly less in the dialyzed patients compared to the control recipients (both *p* < 0.05). The median delay in the onset of systemic reactions and its median duration in dialyzed patients were 1 day and 3 days, respectively. Details are in Figure 1 and Appendix A.

### 3.4. Subgroup Analyses

In univariable analyses, younger patients reported any local reactions (1 and 2 dose) more frequently than older patients (Table 2). In younger patients, some systemic reactions were reported more frequently than in older patients after the second dose of vaccine, i.e., new or worsened muscle pains (20.8% vs. 5.8%; *p* = 0.006), joint pains (20.8% vs. 7.2%; *p* = 0.008), and fever (14.6% vs. 4.3%; *p* = 0.016). Females reported any systemic reactions (2 dose) more frequently than males (Table 2). In details, fatigue (33.8% vs. 16.4%; *p* = 0.009), new or worsened muscle pains (20.0% vs. 4.1%; *p* = 0.004), joint pains (23.1% vs. 4.1%; *p* < 0.001), and chills (7.7% vs. 1.6%; *p* = 0.037) were reported more often in women than in men. Patients with CCI ≥ 6 points reported any local (1 and 2 dose) and any systemic reactions (1 dose) less frequently as compared to subjects with CCI < 6 points (43.7% vs. 80%; *p* < 0.001; 52% vs. 72.6%; *p* = 0.004 and 10.7% vs. 22.4%; *p* = 0.029; respectively). Patients with diabetes reported any local and any systemic reactions less frequently after the first dose of BNT162b2 in comparison with subjects without diabetes (49.3 vs. 66.1%; *p* = 0.022 and 8.5% vs. 20.3%; *p* = 0.03 respectively). Details are in Table 2 and Appendix A.

Four separate regression models were developed to identify independent factors related to reactogenicity (any local reaction for 1 and 2 dose; and any systemic reactions for 1 and 2 dose). Multivariable logistic analysis indicated that age (odds ratio [OR], 0.942; 95% confidence interval [CI]: 0.91–0.976; *p* = 0.001) and female gender (OR, 0.351; 95% CI: 0.184–0.672, *p* = 0.002) were independently associated with reactogenicity of dialyzed patients (age with local reactions after the first dose of BNT162b2 and gender with systemic reactions after the second dose of BNT162b2).

### 3.5. Antipyretic and Pain Medications Use

There were no differences in the use of antipyretic and antipain medications between dialysis patients and age and sex matched controls (5.8% vs. 4.4% after dose 1; 10.7% vs. 13.1% after dose 2) but dialyzed patients used these drugs much less than reported in the pivotal study [5].

### 3.6. Unsolicited Adverse Events

No serious adverse events within 30 min after vaccination were reported by dialyzed patients or controls. Few participants in either group reported unsolicited adverse events up to 30 days after the final vaccination (Table 3). Only shoulder pain reported in a control patient was considered by the investigators to be related to vaccine administration or to the vaccine itself.

### 3.7. Serious Adverse Events

There were 11 serious adverse events reported in dialysis patients in the period up to 30 days after the final vaccination. None of them were considered to be related to BTN162b2. In the dialysis group 5, mild cases of COVID-19 were observed; two of them developed after the first dose of BNT162b2 and this prevented patients from receiving the second dose of the vaccine. Details are summarized in Table 4. In the control group SAE were not noticed.

## 4. Discussion

On 11 December 2020, the U.S. Food and Drug Administration authorized the emergency use of the mRNA vaccine, BNT162b2 from Pfizer-BioN-Tech, against COVID-19 in individuals 16 years of age or older. The biggest vaccination campaign in world history is still underway. The Centers for Disease Control and Prevention (CDC) is conducting ongoing monitoring of COVID-19 vaccine reactivity in people vaccinated in the United States. A report released recently indicates reassuring safety profiles for BNT162b2. Overall, the frequency of reactions reported was in line with the results observed in clinical trials [8]. Quite recently, in a large-scale community-based study in the UK, systemic and local side-effects after BNT162b2 vaccination occur at frequencies even lower than reported in the phase 3 trials [9]. Unfortunately, no dialyzed patients were included in these studies and therefore the safety data of BNT162b2 for this group is lacking. Given the fact that disturbances of acquired immunity in dialyzed (usually elderly) subjects are many and varied, it is uncertain whether vaccination against COVID-19 in them will result in a sufficient immune response, and whether their tolerance to the vaccine is the same as in the general population [10]. Although the risk of adverse events due to over-activation of the immune system is theoretically lower in under such circumstances, this may be offset by an increased predisposition to adverse events overall in patients characterized largely by high frailty [11]. Our study is one of the first to raise this issue [12].

No unexpected patterns of concern were identified in our dialyzed patients. Mild-to-moderate injection site pain was the most common reaction after both the first and the second dose of the vaccine. The most common systemic reactions were fatigue, myalgia, and joint pains, and they were more frequently reported after the second dose of vaccine. Of importance, many local and systemic solicited side-effects were observed less frequently in dialyzed patients than in the age and sex matched control group and much less frequently than reported in the pivotal study including over 10% more women and substantially younger patients than our control group [5]. For instance, in the phase 3 clinical trials of the BNT162b2 vaccine any solicited systemic reactions were reported in 59.1% (dose 1) and 69.9% (dose 2) subjects, which is almost four times and over twice more frequent than in our dialysis group, respectively. Similarly, the use of antipyretic and pain medications reported in the pivotal trial by 24.3% (dose 1) and 41.8% (dose 2) was about four times more common than among our dialysis patients [5].

Reactogenicity was generally mild or moderate, side-effects were transient and resolving in most participants by day three after vaccination, without sequelae. Similar to the control group and general population studies, younger patients and women were more likely to report adverse effects than older subjects and men, respectively [9,13]. Interestingly, younger people and women also appear to show a stronger immune response after vaccination, as we demonstrated in our recent cross-sectional study in hemodialyzed patients [14]. Differences between women and men may therefore be the result of biological differences but also due to inconsistent reporting, i.e., women may exhibit a greater immune response to vaccines than men and experience more side-effects, but men may also be reporting them less frequently. Moreover, sex hormones influence immune responses and cytokine levels, with high doses of oestrogens and androgens having immunosuppressive effects. In addition, the physiological functions of the immune systems evolve throughout life. Reporting rates of adverse events decreases in adulthood, most likely due to a greater tolerance to pain and disease symptoms acquired with life experience and/or a weakening of innate immune defense mechanisms. The latter is supported by the fact that older adults exhibit lower systemic concentrations of IL-10, IL-6, and CRP after vaccination, which may contribute to their tendency to report fewer systemic adverse events. The median age of the studied population was 68 years. Additionally, in maintenance dialysis patients uremia-associated immunodeficiency, immune system senescence and accelerated “inflammaging“ are observed. All these may in part be responsible for the low reactogenicity observed in the studied group [15,16].

The frequency and severity of solicited reactions was not affected by obesity, smoking habit, dialysis modality, and its duration. Although there were also differences between some strata of comorbidities, the study was not powered enough to consider them as truly relevant. Moreover, these associations were not confirmed in multivariable regression analysis. Unsolicited adverse events reported by the dialyzed patients were rare and appear to not be related with the vaccination. Unrelated serious adverse events observed among 11 dialysis patients most likely result from high comorbidity and the high frequency of different dialysis-specific complications. The nonfatal breakthrough infection with SARS-CoV-2 observed in five dialyzed patients requires the attention to and analysis of vaccination effectiveness in this population with future studies. Although rare, breakthrough infection can occur because vaccines against SARS-CoV-2 do not offer 100% protection according to the pivotal studies [5]. According to the California Department of Health, as of 2 June, there had been 5723 breakthrough cases among more than 17.5 million fully vaccinated residents, for a rate of 0.032%. Similar to our report, the majority of the cases were asymptomatic, and the rest had a mild or moderate course. In our study, the incidence rate among those who received at least one dose of the vaccine was 2.6% and 1.6% among the fully vaccinated with two doses. It should be noted that the breakthrough rate reported by the national surveillance in various countries is most likely underestimated because it relies on passive and voluntary reporting, and data might not be complete or representative. In the recent study among the health care workers who received both doses and completed at least 14 days of follow-up after the second dose, the incidence of breakthrough infection was similar to ours and amounted to 1.6% (48 of 3000 health care workers) [17].

Finally, it must be noted that observational study designs have some inherent limitations. For example, the control group was not matched for the most common comorbidities such as diabetes and hypertension. The impact of media news on over-reporting may not be excluded. On the other hand, patients receiving routine care medications in observational studies create a different level of awareness of side effects than participating in an experimental trial and generally tend to report fewer side events. The study sample was not sufficiently large to reliably detect uncommon adverse events and to distinguish vaccine immunogenicity in specific patient subgroups. We only assessed the short-term adverse effects. This population should be further monitored to investigate possible future effects.

## 5. Conclusions

In this study, we have shown that BNT162b2, an mRNA vaccine from Pfizer-BioNTech against SARS-COV-2, is safe and well-tolerated by dialyzed patients. Local and systemic side-effects are mild to moderate in severity, short-lived and less frequent than those observed in the non-dialyzed population. The results can be useful for the nephrological community to resolve patients’ doubts and reduce their vaccine hesitancy.

## Figures and Tables

**Figure 1 medicina-57-00732-f001:**
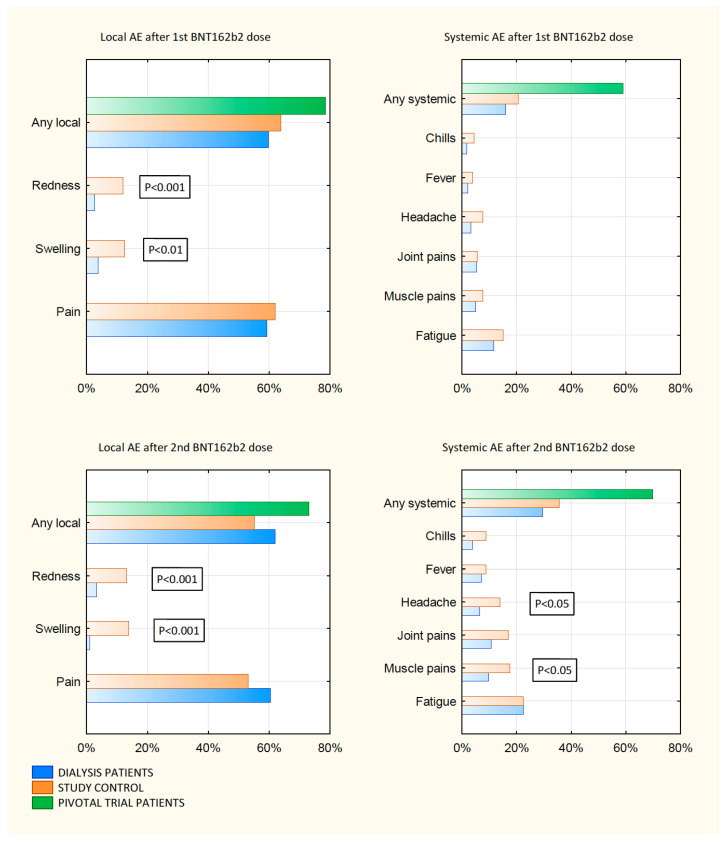
Local and systemic solicited adverse events after 1st and 2nd vaccine dose.

**Table 1 medicina-57-00732-t001:** Characteristics of the study and the control group.

	Study Group	Control Group
	*N* = 190	*N* = 160
Age years median (IQR)	68 (55–74)	63 (53–77) ^a^
Male gender *n* (%)	123 (64.7)	97 (60.6) ^a^
BMI kg/m^2^ median (IQR)	25.5 (22.3–28.9)	26.5 (24.2–29.5) ^b^
Duration of RRT months median (IQR)	36 (14–67)	NA
HD *n* (%)	169 (89.0)	NA
PD *n* (%)	21 (11.0)	NA
Primary nephropathy		
Diabetic nephropathy	46 (24.2)	NA
Glomerulonephritis	23 (12.1)	NA
Polycystic kidney disease	11 (5.8)	NA
Hypertensive nephropathy	10 (5.3)	NA
Unknown	97 (51.1)	NA
Diabetes mellitus *n* (%)	72 (37.9)	25 (16.6) ^c^
Neoplastic disease *n* (%)	28 (14.7)	15 (9.4) ^a^
Arterial hypertension *n* (%)	144 (75.8)	66 (41.3) ^c^
Charlson Comorbidity Index median (IQR)	6 (4–8)	2 (1–4) ^c^
Smokers *n* (%)	28 (14.7)	16 (10.0) ^a^

HD, hemodialyzed patients; PD, dialyzed with peritoneal dialysis patients; RRT, renal replacement therapy; NA, not applicable. Significance (study group vs. control group): ^a^—non significant; ^b^—*p* = 0.033; ^c^—*p* < 0.001.

**Table 2 medicina-57-00732-t002:** Local and systemic solicited adverse events in dialyzed patients in univariable strata analyses.

Subgroups	N1- 1st Dose N2- 2nd Dose	Any Local 1 *n* (%)	Any Local 2 *n* (%)	Any Systemic 1 *n* (%)	Any Systemic 2 *n* (%)
Age < 55	N_1_ = 48; N_2_ = 48	43 (89.6) ^a^	39 (81.3) ^d^	10 (20.8)	18 (37.5)
Age > 55	N_1_ = 141; N_2_ = 139	70 (49.6)	77 (55.4)	20 (14.2)	37 (26.6)
Female	N_1_ = 66; N_2_ = 65	43 (65.2)	46(70.8)	15 (22.7)	29 (44.6) ^a^
Male	N_1_ = 123; N_2_ = 122	70 (56.9)	70 (57.4)	15 (12.2)	26 (21.3)
Diabetes (+)	N_1_ = 71; N_2_ = 70	35 (49.3) ^f^	40 (57.1)	6 (8.5) ^g^	18 (25.7)
Diabetes (-)	N_1_ = 118; N_2_ = 117	78 (66.1)	76 (65.0)	24 (20.3)	37 (31.6)
Neoplastic disease	N_1_ = 28; N_2_ = 28	11 (39.3) ^e^	14 (50.0)	2 (7.1)	7 (25.0)
Neoplastic disease (-)	N_1_ = 161; N_2_ = 159	102 (63.4)	102 (64.2)	28 (17.4)	48 (30.2)
HD modality	N_1_ = 169; N_2_ = 167	97 (57.4)	104 (62.3)	24 (14.2)	48 (28.7)
PD modality	N_1_ = 20; N_2_ = 20	16 (80.0)	12 (60.0)	6 (30.0)	7 (35.0)
CCI < 6	N_1_ = 85; N_2_ = 84	68 (80.0) ^a^	61 (72.6) ^b^	19 (22.4) ^c^	26 (31.0)
CCI ≥ 6	N_1_ = 103; N_2_ = 102	45 (43.7)	53 (52.0)	11 (10.7)	29 (28.4)
BMI < 25.5	N_1_ = 95; N_2_ = 94	61 (64.2)	58 (61.7)	17 (17.9)	32 (34.0)
BMI ≥ 25.5	N_1_ = 94; N_2_ = 93	52 (55.3)	58 (62.4)	13 (13.8)	23 (24.7)
Dialysis vintage < 36 months	N_1_ = 90; N_2_ = 90	54 (60.0)	50 (55.6)	12 (13.3)	22 (24.4)
Dialysis vintage ≥ 36 months	N_1_ = 99; N_2_ = 97	59 (59.6)	66 (68.0)	18 (18.2)	33 (34.0)
Smokers and past-smokers	N_1_ = 111; N_2_ = 109	63 (56.8)	63 (57.8)	13 (11.7)	31 (28.4)
Nonsmokers	N_1_ = 78; N_2_ = 78	50 (64.1)	53 (67.9)	17 (21.8)	24 (30.8)

**Legend:** HD—hemodialysis; PD—peritoneal dialysis; CCI- Charlson Comorbidity Index; BMI—body mass index. Significance: ^a^—*p* < 0.001; ^b^—*p* = 0.004; ^c^—*p* = 0.029; ^d^—*p* = 0.001; ^e^—*p* = 0.016; ^f^—*p* = 0.022; ^g^—*p* = 0.03; grey fields indicate statistically significant differences.

**Table 3 medicina-57-00732-t003:** Unsolicited adverse events and serious adverse events.

	Unsolicited Adverse Events (*n*)	Serious Adverse Events (SAE) (*n*)
Dialyzed patients	Increased sweating (2) Changes in the sense of smell and taste (2) Supraventricular arrhythmias (2)	COVID-19 after first dose (2) COVID-19 after second dose (3) Pneumonia (1) Dialysis peritonitis (1) Catheter-related infection (1) Deterioration of glycemic control (2) Decompensated heart failure (1)
Control patients	Increased sweating (2) Deterioration of glycemic control (1) Shoulder pain (1) Dizziness (2) Sinusitis (1)	Not reported

**Table 4 medicina-57-00732-t004:** Breakthrough SARS-CoV-2 cases among dialyzed study patients.

*n*	5
Age years median (IQR)	74 (66–77)
Sex female/male	1/4
BMI kg/m^2^ median (IQR)	28.0 (22.9–29.3)
Duration of renal replacement therapy months	49 (16.5–81)
Charlson Comorbidity Index (CCI) median (IQR)	8 (4.5–10)
Diabetes mellitus *n* (%)	3 (60.0)
Neoplastic disease *n* (%)	2 (40.0)
Arterial hypertension *n* (%)	5 (100.0)
Primary nephropathy	
Diabetic nephropathy *n* (%)	2 (40.0)
Glomerulonephritis *n* (%)	1 (20.0)
CKD of unknown cause *n* (%)	2 (40.0)
Cases after 1st dose/vaccinated patients *n* (%)	2/189 (1.06%)
Time interval between vaccination and disease onset days	19 and 21
Cases after 2nd dose/vaccinated patients *n* (%)	3/187 (1.6%)
Time interval between vaccination and disease onset days	3 and 3 and 30
Clinical course *n* (%)	
Asymptomatic	2 (40.0)
Mild without hospitalization	1 (20.0)
Mild with hospitalization	2 (40.0)
Recovery	5 (100.0)

## Data Availability

Detailed data presented in the study are available in Appendix A and on request from corresponding author.

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
