# Peer review of "Safety and Tolerability of the BNT162b2 mRNA COVID-19 Vaccine in Dialyzed Patients. COViNEPH Project"

_medicina, 2021, doi:10.3390/medicina57070732_

Round 1

Reviewer 1 Report

*.In study group, what are the etiologies of ESRD ?

*.In HD patient group, when were they vaccinated ? Before, during or after HD ?

*.Five dialyzed patients have breakthrough infection SARS-CoV-2, what are their clinical characteristics and outcomes ? Also add discussion to compare the percentage of breakthrough infection with other studies.

*.One dialyzed patient developed decompensated heart failure, did the patient have myocarditis ?

Author Response

Dear Editors,

Thank you for reviewing our manuscript. According to the Reviewers recommendations we made some changes in the manuscript. Below we present a summary of the changes made and responses to the comments of Reviewers. Changes made are marked in red in the manuscript.

REVIEWER 1

In study group, what are the etiologies of ESRD ?

Following Reviewer suggestions data on the etiology of ESRD in the study group are entered in Table 1 (page 4).

In HD patient group, when were they vaccinated ? Before, during or after HD ?

HD patients were vaccinated on the day of dialysis after HD session. This information is  provided in the methods and material section (line 63)

Five dialyzed patients have breakthrough infection SARS-CoV-2, what are their clinical characteristics and outcomes ? Also add discussion to compare the percentage of breakthrough infection with other studies.

The characteristics and outcome of 5 breakthrough cases are included in an additional Table 4 (page 9). The discussion was supplemented as suggested by the Reviewer as follows (line 279):

“Although rare, breakthrough infection can occur because vaccines against SARS-CoV-2 do not offer 100% protection according to the pivotal studies. According to the California Department of Health, as of June 2, there had been 5.723 breakthrough cases among more than 17.5 million fully vaccinated residents, for a rate of 0.032 percent. Like our report, the majority of the cases were asymptomatic, the rest had a mild or moderate course. In our study, the incidence rate among those who  received at least one dose of vaccine was 2.6% and 1.6% among fully vaccine with two doses. It should be noted that the breakthrough rate reported by the national surveillance  in various countries is probably underestimated because  relies on passive and voluntary reporting, and data might not be complete or representative. In the recent study among  the health care workers who received both doses and completed at least 14 days of follow-up after the second dose, the incidence of breakthrough infection was similar to ours and amounted 1.6% (48 of 3000 health care workers)”

One dialyzed patient developed decompensated heart failure, did the patient have  myocarditis ?

 Myocarditis was not diagnozed. Heart failure was exacerbated by fluid overload.

Reviewer 2 Report

In this manuscript, Polewska and colleagues explored the safety and tolerability of the BNT162b2 vaccine in a cohort of dialysis patients. Dialysis patients showed mostly mild and transitory reactogenicity. The study is nice and well written, however a few comments may be addressed to it order to improve it.

  • Please, check the text for minor English syntactic mistakes (as an example but not limited to: page 2, line 50, “of patients on dialysis maintenance” should be “of patients on maintenance dialysis”)
  • How did the Authors choose a CCI of 7 to performed the statistical analyses when the median CCI in the dialysis cohort was 7? Please explain or use 7. Similarly, why a threshold of 55 years of age was chosen? Please explain or use 55 years as cutoff in the statistical analyses. Please discuss also the 60 months threshold for the dialysis vintage (median of 36 months in the study cohort).
  • What was the delay between the injection and the onset of symptoms? Similarly, please acknowledge the median duration of the reaction.
  • What was the delay between vaccination and COVID-19 in the 5 patients who developed the disease after the vaccination?
  • Did the Authors test the humoral response by dosing specific anti-Spike protein IgGs after the vaccination? It would be worthwhile including this data in the manuscript.
  • Please add significances in supplementary tables.
  • Could the Authors perform a regression analysis to assess variables associated with an increased reactogenicity in dialyzed patients?
  • Please add the units in table 1, i.e. years, kg/m2, months.
  • Please describe in the Methods who assessed the relationship of adverse events with the vaccination and what criteria were used.
  • Please add a copy of the standardized questionnaire in the supplementary material.
  • Why the control group was not matched also for the most common comorbidities such as diabetes and hypertension?
  • Please show the actual p value for every variable in table 1.

Author Response

Dear Editors,

Thank you for reviewing our manuscript. According to the Reviewers recommendations we made some changes in the manuscript. Below we present a summary of the changes made and responses to the comments of Reviewers. Changes made are marked in red in the manuscript.

REVIEWER  2

  • Please, check the text for minor English syntactic mistakes (as an example but not limited to: page 2, line 50, “of patients on dialysis maintenance” should be “of patients on maintenance dialysis”)

It was corrected. The manuscript was once again checked by native English speaking person

  • How did the Authors choose a CCI of 7 to performed the statistical analyses when the median CCI in the dialysis cohort was 7? Please explain or use 7. Similarly, why a threshold of 55 years of age was chosen? Please explain or use 55 years as cutoff in the statistical analyses. Please discuss also the 60 months threshold for the dialysis vintage (median of 36 months in the study cohort).

In strata analyses, a threshold for age was established the same as in the vaccine registration study (Polack F et al and Walsh E et al)  (i.e. 55 years of age). The thresholds for other nominal variables were chosen arbitrarily. The reviewer's comment prompted us to rerun the subgroup analyses for nominal variables (CCI, BMI, vintage dialysis) using the median as the threshold. This did not change the results and conclusion  of the study, but increased its quality. The relevant information is included in the statistics section (line 88),  and new Table 2 (page 7). Thank you for your valuable attention which allowed us to improve the quality of our report.

  • What was the delay between the injection and the onset of symptoms? Similarly, please acknowledge the median duration of the reaction.

The median delay in the onset of local symptoms and systems was one day. The median duration of local and systemic events was approximately 2.5 and 3 days, respectively. This information is  provided in the results section (line 132 and 149).

  • What was the delay between vaccination and COVID-19 in the 5 patients who developed the disease after the vaccination?

Data was provided in new additional Table 4 (page 9).

  • Did the Authors test the humoral response by dosing specific anti-Spike protein IgGs after the vaccination? It would be worthwhile including this data in the manuscript.

The assessment of the immune response to vaccination was not performed per protocol in the study patients. Only some of our patients had an anti-S antibody control after the second dose of vaccination. Due to incomplete data and the use of various analytical methods they were not reported in the manuscript. Shortly,  the seroconversion in anti-s IgG was achieved in almost all dialyzed patients, but the antibody titers were several times lower than observed usually in people without kidney disease.

  • Please add significances in supplementary tables.

It was done as requested

  • Could the Authors perform a regression analysis to assess variables associated with an increased reactogenicity in dialyzed patients?

It was done as requested. Four separate regression models were developed to identify independent factors related to reactogenicity (any local reaction for 1 and 2 dose; and any systemic reactions for 1 and 2 dose). The analyses are described in methods section (line 99). The results were presented in Results section (subgroup analyses) (line 169).

  • Please add the units in table 1, i.e. years, kg/m2, months.

It was done as requested.

  • Please describe in the Methods who assessed the relationship of adverse events with the vaccination and what criteria were used.

Assessment of causality of adverse events following vaccination was done by two physician using the  WHO algorithm and were finally verified by the senior investigator. It was mentioned in methods and material section (line 85-86)

  • Please add a copy of the standardized questionnaire in the supplementary material.

It was done as requested.

  • Why the control group was not matched also for the most common comorbidities such as diabetes and hypertension?

The study was of an observational nature, which made it extremely difficult to properly match the control group. Efforts were focused on making the groups comparable with age and gender distribution, i.e. the main previously known factors that may influence reactogenicity. It was mentioned in study limitations (line 294).

  • Please show the actual p value for every variable in table 1

It was done as requested. The legend was included to the Table 1

Round 2

Reviewer 2 Report

I have no further comments.